# Potential Role of IL-37 in Atopic Dermatitis

**DOI:** 10.3390/cells12232766

**Published:** 2023-12-04

**Authors:** Alicja Mesjasz, Magdalena Trzeciak, Jolanta Gleń, Marta Jaskulak

**Affiliations:** 1Dermatological Students Scientific Association, Department of Dermatology, Venereology and Allergology, Faculty of Medicine, Medical University of Gdansk, 80-214 Gdansk, Poland; alicja.mesjasz@gumed.edu.pl; 2Department of Dermatology, Venereology and Allergology, Faculty of Medicine, Medical University of Gdansk, 80-214 Gdansk, Poland; jglen@gumed.edu.pl; 3Department of Immunobiology and Environmental Microbiology, Faculty of Health Sciences, Medical University of Gdansk, 80-214 Gdansk, Poland; marta.jaskulak@gumed.edu.pl

**Keywords:** atopic dermatitis, interleukin 37, IL-37

## Abstract

Interleukin 37 (IL-37) is a recently discovered member of the IL-1 cytokine family that appears to have anti-inflammatory and immunosuppressive effects in various diseases. IL-37 acts as a dual-function cytokine, exerting its effect extracellularly by forming a complex with the receptors IL-18 α (IL-18Rα) and IL-1R8 and transmitting anti-inflammatory signals, as well as intracellularly by interacting with Smad3, entering the nucleus, and inhibiting the transcription of pro-inflammatory genes. Consequently, IL-37 is linked to IL-18, which plays a role in the pathogenesis of atopic dermatitis (AD), consistent with our studies. Some isoforms of IL-37 are expressed by keratinocytes, monocytes, and other skin immune cells. IL-37 has been found to modulate the skewed T helper 2 (Th2) inflammation that is fundamental to the pathogenesis of AD. This review provides an up-to-date summary of the function of IL-37 in modulating the immune system and analyses its potential role in the pathogenesis of AD. Moreover, it speculates on IL-37’s hypothetical value as a therapeutic target in the treatment of AD.

## 1. Introduction

Interleukin 37 (IL-37) is a recently discovered anti-inflammatory member of the IL-1 cytokine family [1]. The IL-1 cytokine family is divided into three subfamilies dependent on their structural and functional resemblances: IL-1 (IL-1α, IL-1β, IL-33, and IL-1Ra), IL-18 (IL-18 and IL-37), and IL-36 (IL-36α, IL-36β, IL-36γ, IL-36Ra, and IL-38) [1].

IL-37 was first identified in silico in 2000 and was formerly known as IL-1F7 [2]. Unlike other members of the IL-1 cytokine family, a homologue gene for IL-37 has not been identified in mice; therefore, transgenic mice are used to study this cytokine [2]. Serum concentrations of IL-37 in healthy persons are notably low (100 pg/mL) [2]. However, IL-37 production can be stimulated by pro-inflammatory stimuli, and the cytokine itself serves as a mechanism of self-protection against excessive inflammation and severe tissue destruction, limiting both innate and acquired immunity [3]. Accordingly, IL-37 levels have been found to be elevated in patients with inflammatory and autoimmune disorders [4]. IL-37-transgenic (-tg) mice were protected against a number of animal model diseases, including colitis, acute myocardial infarction, idiopathic pulmonary fibrosis (IPF), obesity-induced inflammation, allergic airway inflammation, and surgical operation-induced spinal cord injury, among others [5,6,7,8]. IL-37 also has great potential in oncology due to its inhibitory effect on the initiation and development of cancer [9]. In addition, IL-37 also plays a role in maintaining homeostasis between host immune responses and microbial species; however, its role in the skin microbiome in atopic dermatitis (AD) has not been thoroughly studied yet [2].

However, IL-37 seems to be involved in the pathogenesis of AD, however, it is unknown to what extent (Table 1). AD is a chronic, inflammatory, pruritic dermatosis that affects approximately 20% of children and 3–7% of adults worldwide, depending on the geographical location [10]. It imposes a significant burden on life for both patients and their parents [10].

As a result of the recent introduction of novel, molecularly targeted therapeutic strategies to patients, global research efforts are concentrated on the identification of additional molecular targets. There exists a limited number of reviews that have explored the involvement of IL-37 in AD [19,20,21,22]. Nevertheless, in the available ones, IL-37 is consistently characterised as an anti-inflammatory agent that alleviates inflammation in AD [19,20,21,22]. Furthermore, they all agree that there is a necessity for more research to comprehensively determine the therapeutic potential of IL-37. It is important to recognise that in asthma, IL-37 possesses the ability to reduce allergic inflammation by targeting not just the Th2 cytokine axis, but also all Th1/Th2/Th17 cytokine axes [21]. Additionally, IL-37 demonstrates a protective effect by modulating atherosclerotic mechanisms, thereby potentially offering significant advantages in the context of cardiovascular comorbidities in AD [21].

All of these suggest that anti-inflammatory IL-37 is an interesting newly recognised player in the broad landscape of cytokines involved in the pathogenesis of AD, and it therefore requires more focus and understanding.

## 2. Discussion

### 2.1. Brief Overview of the IL-37

#### 2.1.1. Production and Processing

The IL-37 gene is located on chromosome 2q12–13 in close proximity to the regulatory areas of the IL-1α and IL-1β genes, which may be important for IL-37’s anti-inflammatory functions, as when pro-inflammatory stimuli induce transcription of the IL-1 genes, IL-37 is also activated [23]. Five distinct isoforms of IL-37 are generated by alternative splicing of IL-37’ messenger ribonucleic acid (mRNA): a–e. [24]. Different IL-37 isoforms are expressed in distinct tissues [25]. IL-37a, b, and c, for example, are predominantly expressed in the thymus, bone marrow, and lymph nodes, among others, whereas only IL-37a is expressed in the brain, and only IL-37c is expressed in the heart [25]. IL-37b is the longest and most thoroughly researched isoform [26]. It contains five of the six exons of the IL-37 gene, excluding exon 3 [24]. IL-37 isoforms a, b, and d share exons 4, 5, and 6 and encode functional proteins fundamental for the establishment of the β-fold barrel structure required for the proper extracellular function of IL-37 [24]. The IL-37 isoforms c and e lack one or more of these exons and may therefore encode non-functional proteins [24]. IL-37, which consists of 12 β-barrel strands, follows the structural pattern of the IL-1 family, but especially that of IL-18 [27]. Caspase-1 has been identified as the primary enzyme needed for the modification of precursors into mature cytokine forms [25,28].

#### 2.1.2. Release

Il-37 is not constitutively expressed in cells from healthy subjects, but its expression is elevated in response to pro-inflammatory stimuli, consistent with the presumption that IL-37 works as a negative feedback system to suppress excessive inflammation [23]. IL-37 is expressed in a number of cells, possibly contributing to the preservation of immunological homeostasis [24]. In resting selected human cells, IL-37 transcript levels are generally very low due to the rapid degradation of IL-37 mRNA caused by instability elements in the coding region [25]. When stimulated with lipopolysaccharide (LPS) or other exogenous stimuli, the stability of IL-37 mRNA is significantly increased [25].

#### 2.1.3. Mechanism of Action

IL-37 serves as a dual-function cytokine that exerts powerful anti-inflammatory effects either intracellularly or extracellularly. It is still unclear which conditions and/or factors govern the selection of one mechanism over another [24].

Not yet identified is the specific receptor for IL-37 [2]. IL-18 has a receptor affinity for IL-18Rα that is 50 times greater than that of IL-37 [29]. IL-37 exhibits binding affinity towards IL-18Rα, consequently facilitating the recruitment of IL-1R8 to establish a receptor complex that conveys an anti-inflammatory signal [2,30]. Therefore, IL-1R8 and IL-18Rα function as IL-37 co-receptors that are required to transmit anti-inflammatory messages (Figure 1) [31]. Interestingly, the extracellular presence of IL-18 binding protein (IL-18BP) has been seen to impede the binding of IL-37 to IL-18Rα [32]. However, IL-18BP has also been found to exhibit a greater affinity for IL-18 compared to IL-18Rα, hence also inhibiting the binding of IL-18 to its receptor [25,32]. The weak inhibitory effect of high doses of IL-37 on the production of inflammatory cytokines may be due to the spontaneous formation of homodimers of IL-37 at high concentrations, which can be perceived as an auto-regulatory mechanism that limits excessive immunosuppression [24].

Providing a brief description of the IL-18 cytokine is important because of its significance within the context of IL-37 [33]. The binding of Pathogen Associated Molecular Patterns (PAMPs) to Toll-like receptors (TLRs) and the subsequent activation of the NF-κB pathway leads to the transcription of the precursor form of IL-18 [33]. This cytokine is synthesised by a number of cell types, including hematopoietic and non-hematopoietic cells such as monocytes, endothelial cells, osteoblasts, and keratinocytes [33]. As it is a potent, pro-inflammatory agent that modulates both innate and adaptive immune responses, IL-18 has frequently been used as a biomarker in numerous studies to assess the activity of inflammasomes [33].

The described interplays are required to activate several intracellular switches that inhibit inflammation. IL-37 leads to the suppression of inflammatory pathways such as mammalian target of rapamycin (mTOR) and mitogen-activated protein kinase (MAPK), as well as NF-κB and various transcription factors [2]. Furthermore, IL-37 leads to activation of signal transducer and activator of transcription (STAT)3, protein phosphatase and tensin homolog (PTEN), and 5′AMP-activated protein kinase (AMPK) [2].

Since IL-37 lacks a nuclear localisation sequence, it is probable that it reaches the nucleus tied to Smad3 and functions by enhancing Smad3′s anti-inflammatory activity rather than via direct DNA binding [34]. In response to pro-inflammatory stimuli, the intracellular concentration of the IL-37 precursor rises, activated caspase-1 cleaves the precursor, and the C-terminal domain of IL-37 binds to Smad3 [2]. This complex relocates to the nucleus after being phosphorylated, where it takes part in the gene expression regulation process [2].

### 2.2. IL-37 and Immune Cells

#### 2.2.1. Monocytes and Macrophages

The Toll-like receptor (TLR) ligands LPS and Pam3CysSerLys4 (Pam3CSK4) as well as transforming growth factor (TGF)-β1 induce IL-37 production [35]. In addition, IL-18, IFN-γ, IL-1β, tumour necrosis factor (TNF), and the dinucleotide CpG also stimulate the synthesis of IL-37, whereas the combination of Granulocyte-Macrophage Colony-Stimulating Factor (GM-CSF) and IL-4, a key cytokine engaged in the pathogenesis of AD, inhibit IL-37 expression [35]. IL-37 appears to be stored in monocytes for fast release and can be released as early as 3 h after LPS stimulation [36].

IL-37 inhibits the expression of intracellular adhesion molecule 1 (ICAM-1), Macrophage Colony-Stimulating Factor (M-CSF), GM-CSF, pro-inflammatory cytokines (IL-1α-by 88%, IL-1β, IL-1Ra, IL-6 by 86%, IL-8, IL-23, TNFα), chemokines (C-X-C Motif Chemokine Ligand (CXCL)2 (MIP-2), Chemokine C-C motif ligand 12 (CCL12), and CXCL13 (BCA-1)) [31,35,37]. Therefore, IL-37 inhibits the proliferation, apoptosis, and transmigration of macrophages [38].

Macrophages exhibit two major functional phenotypes: the classical activation phenotype, also known as M1 macrophages, and the alternative activation phenotype, also recognised as M2 macrophages [39]. M1 macrophages induced by TLR ligands and IFN-γ are primarily responsible for pro-inflammatory responses, whereas M2 macrophages induced by IL4/IL13 (M2a), immune complexes (M2b), and IL-10 or TGF-β (M2c) are principally in charge of anti-inflammatory responses [39]. IL-37 inhibits Notch1 and N*F-k*B activation, the pathways that regulate macrophage polarisation, to attenuate M1 polarisation [39]. Furthermore, when exposed to recombinant IL-37, M2 macrophages and their anti-inflammatory cytokine secretion are enhanced [39].

#### 2.2.2. Dendritic Cells (DCs)

DCs secrete IL-37 in response to LPS stimulation as well as at a steady state, contributing to the maintenance of an anti-inflammatory state [36]. Induction of IL-37 expression in DCs unfavourably modulates DC maturation and function, thus inducing semimature tolerogenic DCs that impede T effector (Teff) cell activation and promote the development of regulatory T (Treg) cells [40]. IL-37 might inhibit DC maturation via the IL-1R8–TLR4–N*F-k*B pathway [41]. Semimature tolerogenic DCs express lower amounts of major histocompatibility complex (MHC) II and costimulatory molecules; release fewer pro-inflammatory cytokines than mature DCs, including L-1, IL-6, and IL-12; and produce more immunosuppressive mediators such as IL-10, thereby further enhancing the expansion of Tregs [40].

### 2.3. Pathogenesis of AD

The aetiology of AD involves multiple factors. The root causes include a combination of genetic and environmental factors, abnormalities in the immune system’s response, defects in the integrity of the epidermal barrier, and the imbalance of the skin’s microbial arrangement [42]. The pathophysiology of the first three elements will be briefly discussed in this section, while the remaining factors are going to be covered in the sections to come.

Multiple genes have been identified as potentially related to AD and they may be categorised into five primary groups: epidermal barrier genes (the filaggrin gene mutation being the most well known), genes of innate and adaptive immune mechanisms, genes encoding DNA methylation, genes encoding vitamin D metabolism, as well as genes encoding alarmins produced by keratinocytes [43]. Currently, there is a lack of research related to the relationship between IL-37 gene variants and AD. Various environmental factors, such as the adoption of a Western lifestyle, exposure to air pollution, increasing rates of obesity, escalated use of antibiotics, as well as smoking, are believed to play a role in the development of AD, likely via epigenetic mechanisms [42,43].

AD is primarily a Th2-mediated dermatosis; however, it is unknown whether the immune disruption itself or the disrupted, leaking epidermal barrier enabling the allergens and pollution to transfer is the initial root cause of this disease [42]. The overproduction of Th2 cells results in an elevated production of the cytokines IL-4, IL-5, and IL-13 that, in turn, stimulate the production of IgE and eosinophils as well as lead to a reduction in the expression of filaggrin in the epidermis [42]. However, in acute lesions, cytokines from the Th2 and Th22 axes, as well as Th17 to a lesser degree, are elevated [42]. Th2 and Th22 responses are amplified with the chronicity of the disease process, which is accompanied by a rise in Th1 response biomarkers, which produce TNFα, and INF cytokines, among others [42]. Tregs are essential for immune suppression including the control of allergic responses [44]. However, their role in AD is still not entirely understood [44].

It is important to mention that the phenotype and endotype of AD vary by population. Asian AD is distinguished immunologically from European/American AD by a greater activation of the Th17 axis and a diminished Th1 response [10]. Th1/Th17 attenuation and Th2/Th22 skewing characterise African/American AD, which is similar to paediatric AD, in which the Th2 response is more prominent and associated with the atopic march, whereas the Th1 response is diminished [10].

### 2.4. IL-37 and Immune Response in AD

#### 2.4.1. T Helper (Th) Cells

IL-37 appears to be decreased in the serum of AD patients [17,45], but not across all studies [46]. Additionally, Thijs et al. observed decreased serum levels of IL-37 to be indicative of a more severe AD endotype, which was confirmed by Fujita et al. [46,47]. Furthermore, IL-37 seems to be downregulated in the skin lesions of AD patients [11,13,15,48]. Moreover, it has been observed that IL-37 is downregulated in both asthma and allergic rhinitis. This suggests that there may be a potential deficiency in the production of IL-37 in individuals with AD, as well as in other disorders characterised by allergic inflammation [6,49]. Downregulation of IL-37 in allergic patients leads to ineffective immune suppression and dysregulations in response to allergen stimulation [17]. Figure 2 illustrates the role of IL-37 in AD. The secretion of IL-37 by peripheral blood mononuclear cells (PBMCs) exhibited a notable decrease subsequent to stimulation with recombinant Th2 cytokines [28]. Conversely, there was no observable alteration in IL-37 production following stimulation with Th1 cytokines [28]. This observation implies that the Th2 immune response has the capacity to inhibit the production of IL-37 [28].

In a mouse model of experimental allergic asthma, local application of recombinant IL-37 lowered the secretion of IL-4, IL-5, and IL-13, thereby reducing Th2-mediated allergic airway inflammation [50]. In a murine model of allergic rhinitis (AR), administration of IL-37 decreased the number of eosinophils in the nasal mucosa, restored its thickness, lowered the frequency of nasal rubbing and sneezing, and decreased the levels of immunoglobulin (Ig)E, IgG1, IgG2a, IL-4, IL-13, IL-17a, and C-C Motif Chemokine Ligand 11 (CCL11), which recruits eosinophils [51]. In a murine model of invasive pulmonary aspergillosis, IL-37 significantly inhibited the recruitment of inflammatory cells and the activation of Th2 [52]. According to a study performed by Hou et al., IL-37b decreased the expression of the thymic stromal lymphopoietin (TSLP) receptor on basophils, decreasing the activation of basophils, therefore inhibiting Th2 inflammation as activated basophils secrete the Th2 cytokine IL-4, which causes the differentiation of naïve T cells into Th2 cells for the Th2 immune response [12]. STAT6 activation, which requires IL-4, transforms CD4+ T cells into Th2 cells by upregulating GATA3 [53]. STAT6 phosphorylation and IL-4 levels were both reduced in IL-37-treated animals, indicating that IL-37 may downregulate GATA3 expression by blocking the IL-4/STAT6 signalling pathway, thereby decreasing the proliferation, differentiation, and activation of Th2 cells in the AR model of IL-37-tg mice [53]. In contrast, in the study conducted by Lv et al., IL-37 treatment did not appear to affect Th2 cell differentiation, recruitment, or activation in the ovalbumin (OVA) model house dust mite (HDM)-induced asthma model [54]. IL-4 and IL-13 were found to inhibit caspase-1 activity in nasal epithelial cells, which may contribute to a decrease in the secretion of IL-37 [55].

In vivo studies have shown that IL-37 can block the activity of not only Th2, but also Th1 and Th17 cells via PBMCs, M1 macrophages, and DCs [27]. Human DCs treated with IL-37 acquired a tolerogenic phenotype that declines Th1 and Th17 populations [30]. IL-6 is essential for STAT3 activation, and thereby proliferation, differentiation, and activation of Th17 cells [53]. IL-6 was observed to be downregulated in the IL-37-tg AR mouse model [53]. Th1- and Th17-polarising conditions stimulated the stable expression of IL-37 in CD4+ T cells, whereas Th2-polarising conditions had no effect [56]. Th2-secreted cytokines IL-4 and GM-CSF suppressed constitutive IL-37 expression, whereas the Th1-secreted cytokine IFN-γ and Th-17-derived IL-17 were very effective at inducing IL-37 [56]. Th1/Th17 cytokines (IFN-γ and IL-17) may influence IL-37 production in CD4+ T cells, whereas the Th2 cytokine IL-4 may inhibit IL-37 production in CD4+ T cells [56].

#### 2.4.2. Regulatory T Cells (Tregs)

Regulatory T cells (Tregs) are essential for maintaining immune tolerance through the suppression of T effector (Teff) cells’ action and proliferation [44]. Tregs are essential in the maintenance of self-tolerance and the prevention of allergic responses and other immune reactions [57]. Their blood count is positively correlated with the severity of AD; however, their role in this disease is still poorly understood [57].

IL-37 maintained a low level of expression in freshly isolated human CD4+CD25+ Tregs that were not stimulated [3,58]. Silencing IL-37 in human CD4+CD25+ Tregs clearly lowered their suppressive activity [58]. In a study by Wang et al., IL-37 stimulation significantly increased the suppressive activity of CD4+CD25+ Tregs via upregulation of CTLA-4 and FOXP3, and led to an increase in TGF-1, but IL-10 levels did not rise [59]. TGF-β and IL-10, and expression of FOXP3 and CTLA-4, declined significantly when the IL-37 gene was silenced by siRNA in the study by Shuai et al. [58]. Clearly, IL-37 can boost Treg cell activity and contribute to their production of TGF-β and/or IL-10 which, consequently, contribute to Teff cell inhibition [3,11,58,59]. Interestingly, the considerable rise in the IFN-γ/IL-4 ratio in Tregs once IL-37 expression was silenced by siRNA implies that IL-37 may be associated with the process of CD4+CD25+ Treg-induced type 2 T-cell polarisation [58].

IL-2 affects fundamental aspects of the immune response and homeostasis [60]. Dual opposing functions are served by IL-2; it amplifies proliferative responses of Teff cells and natural killer (NK) cells while regulating immune homeostasis by driving Treg cell proliferation, differentiation, and function [60]. When the concentration of IL-2 is small, Treg cells respond predominantly [60]. Doses that stimulate the activity of Teff cells and NK cells simultaneously induce Treg cell responses, consequently exacerbating inflammation [60]. In a study performed by Wang et al., there was no difference in IL-2 levels between the IL-37-stimulated group and the control group [59]. In contrast, in the study conducted by Shuai et al., the downregulation of IL-37 levels appeared to be responsible for the increase in IL-2 secretion, resulting in exacerbation of inflammation [58].

Primary bile cholic acids can be converted into secondary bile acids using microbial bile salt hydrolases, which are predominantly expressed by Lactobacillus and Bifidobacteria [11]. Secondary bile acids inhibit the NLRP3 inflammasome, and this is associated with the suppression of inflammation and an increase in Tregs and IL-10 [11]. Primary bile acid and Lactobacillus were considerably increased in IL-37b-tg AD mice [11].

#### 2.4.3. Eosinophils

Eosinophils, together with basophils and mast cells, are essential components in the pathogenesis of allergic inflammation [61]. IL-5 and GM-CSF, which are secreted by lymphocytes and mast cells, contribute to the regulation of eosinophil production, maturation, and differentiation [61]. CCL11 is a chemokine that facilitates the targeted recruitment of eosinophils [51,54,61]. It is synthesised by several cell types, such as lymphocytes, macrophages, and fibroblasts [51,54,61]. The stimulation of eosinophils by Th2 cytokines can result in the production of IL-12 [61]. Consequently, eosinophils have the potential to facilitate a transition from a Th2-like immune response observed in acute lesions to a Th1-like immune response observed in chronic lesions of AD [61]. Even though elevated eosinophil levels are risk factors characteristic of AD, elevated eosinophil blood counts are not observed in all AD patients [62].

Numerous studies have demonstrated that IL-37 reduces eosinophilic inflammation [12,53,59]. In a study performed by Hou et al., IL-37 additionally decreased eosinophilic infiltration in AD skin-like lesions [11].

Autophagy plays an important function in controlling the development of the immune system, activating the host’s adaptive and innate immune responses [11]. It is reliant on the regulation of the intracellular 5’AMP-activated protein kinase (AMPK)–mTOR signalling pathway [11]. The autophagy inhibitor *3-methyladenine* (3-MA) was significantly decreased, whereas AMP, the activator of AMPK for autophagy induction, was significantly increased in IL-37b-tg AD mice, showing that IL-37 could enhance autophagy in eosinophil-mediated allergic inflammation [11].

IL-37 was found to inhibit the expression of CCL11 [51,54]. IL-18, which is elevated in AD patients, enhances human invariant NK (iNK) cells and endothelial cells and induces the eosinophil-activating cytokines IL-5 and IL-13 [17,63].

TSLP was discovered to induce chemotactic and pro-survival functions in eosinophils. IL-37 suppresses TSLP, thereby inhibiting these processes [12].

The expression of Mex3 RNA-binding family member B (Mex3B), which is an RNA-binding protein and coreceptor of Toll-like receptor (TLR) 3, was downregulated by IL-37. Consequently, IL-37 suppressed the synthesis of thymic stromal lymphopoietin (TSLP) in nasal epithelial cells, which consequently alleviated eosinophilic inflammation in an in vivo mouse model [55]. There is a body of evidence indicating that TSLP can activate DCs and mast cells, leading to the induction of Th2-type immune responses, and it is increasingly suggested that TSLP may be considered an important molecular player in the pathophysiology of AD.

#### 2.4.4. Basophils

After allergen contact, epithelial cells produce inflammatory mediators such as TSLP that recruit and activate basophils [64]. The complete nature of basophils’ involvement in AD remains little explored [65]. Although these cells constitute a small proportion of cellular infiltrates in skin lesions, they appear to have a role in promoting a shift towards Th2 immune responses by releasing IL-4, which stimulates the differentiation of naïve T cells into Th2 cells as well as contributes to the development of pruritus in AD [12,65,66].

In basophils, IL-37b inhibited the expression of the TSLP receptor [12]. Nonetheless, heterogeneous IL-37-tg mice exhibited no significant differences in basophil populations or AD symptoms, nor scratching behaviour, compared to wild mice [12]. However, IL-37b intraperitoneal treatment significantly decreased the basophil infiltration in the ear and spleen, the Th2 immune response, and scratching times in AD mice compared to control mice [12]. This may be because IL-37 heterozygous transgenic mice express insufficient amounts of IL-37b [12]. It has been demonstrated that IL-37 increases TLR9 mRNA expression in basophils [67]. A low level of TLR9 expression in the nasal mucosa of AR patients appeared to be associated with Th2-type allergic inflammation [67].

#### 2.4.5. Mast Cells

Mast cells (MCs) can phagocytose and kill bacteria, process protein antigens, function as antigen-presenting cells (APCs), and produce a variety of mediators, including those belonging to the IL-1 cytokine family [68]. The modulation of naïve T cell differentiation into Th1 and Th2 subsets, together with the enhancement of T-cell activation, regulation of primary B-cell formation, and stimulation of IgE synthesis in B cells, are also among their functions [61]. Mast cells play a crucial role in shifting the T cell response towards the Th2 axis in AD by inhibiting the production of IL-12 by dendritic cells [61].

It is possible to induce the inflammatory activity of MCs without the participation of IgE [69]. IL-33 seems to be a powerful stimulator of MCs, inducing migration, maturation, and adhesion, as well as promoting survival and the production of a variety of pro-inflammatory cytokines in these cells [70]. Some evidence suggests that the regulation of the NF-κB and P38 MAPK signalling pathways is associated with IL-33-induced mast cell inflammation [69]. IL-37 inhibited the activation of NF-κB and the phosphorylation of P38 MAPK with the participation of Smad3 [69]. These findings may indicate that IL-37 plays a role in reducing mast cell inflammation [69].

Extracellular IL-37 is active as a monomer, whereas binding to heparin promotes its homodimerisation, with IL-37 dimers inhibiting the monomer’s activity [71]. The richest source of heparin is mast cells, and heparin inhibits the action of IL-37 by promoting the formation of homodimers [71]. This may imply that compounds that preferentially inhibit the secretion of mast cell mediators can be used in conjunction with IL-37 as new therapeutic agents [24].

#### 2.4.6. B Cells

Activated B cells undergo a process known as immunoglobulin class-switch recombination, which is regulated by the signalling of IL-4 and IL-13, which are dependent on the presence of IL-4Rα and STAT6 [72]. This leads to a switch from the synthesis of IgM antibodies to the generation of IgE ones, which has a major role in promoting allergic inflammation [72]. In a murine model of AR, the administration of IL-37 resulted in a notable reduction in the levels of IL-4 and IL-13 in both the serum and nasal lavage fluid [51]. This subsequently led to a decrease in IgE levels [51]. However, it is important to note that a number of studies have reported a lack of statistically significant association between IL-37 and IgE levels specifically in AD [17,28,73].

### 2.5. IL-37 and Skin Barrier Disruption

In addition to filaggrin (FLG), the expression of loricrin (LOR), involucrin (IVL), and FLG2 is downregulated in the lesional and nonlesional skin of AD, resulting in an abnormal composition of the corneocyte lipid envelope surrounding corneocytes and, therefore, an impaired epidermal barrier [74]. As a result, there is an elevation in passive water loss from the body, a rise in the absorption of pollutants and allergens from the external environment, and an increased susceptibility to microbial infection [75]. An elevation in pH levels, together with heightened serine proteinase activity, as well as increased levels of cytokines like IFN-α, contribute to the inactivation and destruction of enzymes that are essential for the synthesis of ceramides [75]. Furthermore, not only the chain lengths of ceramide are shortened, but also those of free fatty acids and esterified fatty acids [75]. In the context of AD, numerous alterations emerge in the physical, chemical, and antimicrobial properties of the skin, resulting in a reduction in the diversity of the skin microbiota and increased colonisation by pathogens, notably *Staphylococcus aureus* [76]. Regrettably, the current body of research on the relationship between the skin microbiota in AD and IL-37 is limited.

A decrease in IL-37 protein expression was observed in the epidermis of AD patients. [13,15,17], but not across all studies [46]. The greatest decrease in IL-37 was found in chronic AD skin lesions [15]. IL-37 production has been reported in epidermal keratinocytes that are more mature and differentiated [15,77].

Zhou et al. analysed the relationship between IL-37 and the expression of FLG, FLG2, and IVL and established a positive correlation [15]. In the EpidermFT in vitro 3D human skin model, IL-4, IL-13, and IL-31 were enough to decrease IL-37, whereas IL-37 increased FLG and FLG2 but did not impact the expression of IVL, indicating that IL-37 may preferentially affect FLG and FLG2, but not all epidermal differentiation complexes (EDCs) [15]. This was not in line with the study conducted by Tianheng et al., in which IL-37 was strongly negatively associated with IVL expression [17]. The concentration of IL-37 in the epidermis was negatively correlated with 39 of 41 pro-inflammatory proteins [17]. A reduced epidermal thickness and a considerably lowered scratching frequency were observed in IL-37-tg mice [11].

Il-37 was found to inhibit, presumably by inhibiting MAPK and STAT1 activation in keratinocytes, the expression of keratinocyte-derived IL-33 [14,78]. IL-33 induces a type 2 immune response; directly stimulates nerves, resulting in pruritus; and has a well-established role in the pathogenesis of AD [14]. All of these findings indicate that IL-37 may modulate epidermal barrier function [15]. Nevertheless, it is crucial to note that a significant percentage of those diagnosed with AD have an inherent mutation in the FLG gene. Consequently, it may be concluded that the administration of IL-37 alone is insufficient for these patients.

### 2.6. IL-37 as a Potential Therapeutic Target in AD

With the increasing availability of biological drugs and Janus kinase inhibitors, there is a significant emphasis on the search for additional therapies that directly modulate the inflammatory processes and precisely target immune pathways or agents involved in the pathogenesis of AD. Possibly in the future, AD will be treated with therapies that directly stimulate cells to secrete anti-inflammatory cytokines such as IL-37 or that are based on the administration of recombinant anti-inflammatory cytokines.

IL-37 is sometimes referred to as a ‘peacemaker’ because it is emerging to be a very powerful anti-inflammatory target [79]. Consequently, to prevent excessive silencing of innate and adaptive immunity, IL-37 function must be tightly controlled, provided that it will be considered for use in therapy in the future [79]. Physiologically, it is possible to achieve, for instance, through the instability of mRNA and homodimerisation [79]. The question is whether such regulation is achievable through the implementation of exogenous, recombinant IL-37 [79]. Furthermore, human IL-37 therapeutic benefits are derived from acute disease models, with the exception of a few chronic conditions [79]. This raises the question of whether IL-37 can even be considered as a potential treatment for AD. It might prove most useful for treating exacerbations, but these are merely speculations. However, we are at the point where IL-37 has been administered successfully in animal models of allergic diseases [6,50,51]. Hypothetical future in vivo evaluations on humans will undoubtedly provide new information regarding the “natural conditions” in which IL-37 functions within the human body. At this point, it is clear that more observational studies looking at the levels and changes in IL-37 in response to different medications, at different stages of AD, and in patients who have comorbid conditions could be beneficial.

As a matter of fact, there is hope for the use of IL-37 in a large variety of other conditions, including autoimmune, neurological, cardiovascular, and neoplastic diseases, to name but a few [80,81,82,83].

## 3. Conclusions

IL-37 appears to have a significant anti-inflammatory effect on modulating the inflammatory response, particularly during inflammation. The role of IL-37 in the pathogenesis of AD through various immunological mechanisms appears to be a fact, but the extent of this role is unknown and requires additional research. In order to determine the potential therapeutic potential of IL-37 in the future, it is necessary to conduct additional studies evaluating the pathways that are triggered or inhibited by IL-37 to gain an in-depth understanding of this cytokine in order to develop safer and more precise therapies.

## Figures and Tables

**Figure 1 cells-12-02766-f001:**
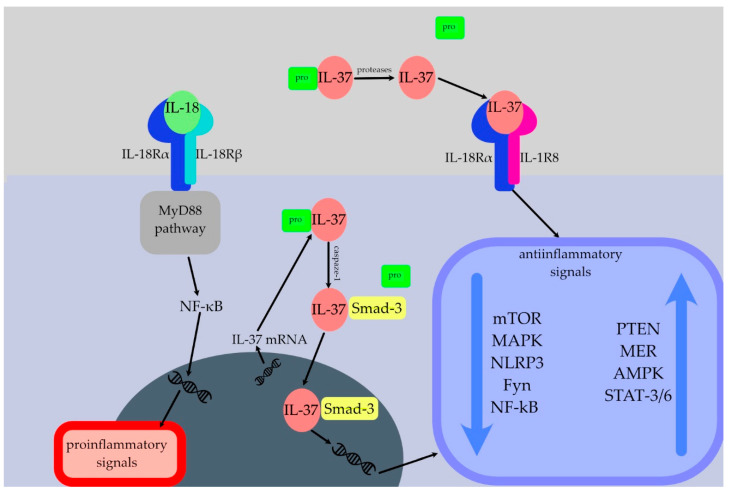
Interleukin (IL)-37 maintains its anti-inflammatory outcomes by limiting IL-18Rα from adhering to IL-18 and by attracting the orphan decoy IL-1R8 to the IL-18R/IL-37/IL-1R8 complex. This prevents IL-18Rα from participating in the transduction of pro-inflammatory IL-18 signals via the myeloid differentiation primary response 88 (MyD88) pathway and NF-κB. Focal adhesion kinase (FAK), signal transducer and activator of transcription (STAT)1, STAT3, mammalian target of rapamycin (mTOR), p53, p38, paxillin, spleen tyrosine kinase (Syk), Src homology region 2-containing protein tyrosine phosphatase 2 (SHP-2), and protein kinase B (AKT) are pro-inflammatory signal-transducing mediators whose activation is reduced by IL-37, whereas protein phosphatase and tensin homolog (PTEN), which inhibits the PK3 kinase, FAK, mTOR, and mitogen-activated protein kinase (MAPK) pathways is enhanced by IL-37. mTOR has a pro-inflammatory function as it blocks IL-10 and promotes IL-12, as well as an anti-inflammatory function as IL-6 is blocked by rapamycin. MAPK regulates the production of pro-inflammatory cytokines such as IL-6 and tumour necrosis factor (TNF)-α. After activation of NLRP3, there is a subsequent formulation of the NLRP3 inflammasome that cleaves, for instance, pro-inflammatory IL-18 to its active form. Inhibition of Fyn dampens the systemic inflammation and Fyn is engaged in pro-inflammatory stimuli, for instance by stimulating the production of IL-6 and TNF-α. NF-κB is central in the regulation of inflammatory processes and stimulates the production of many pro-inflammatory cytokines including IL-1, IL-12, IL-18, and chemokines such as CXCL10. PTEN negatively regulates the expression of pro-inflammatory cytokines such as IL-1, IL-6, and TNF- α. Mer decreases the levels of CCL1 and IL-6. AMPK as well as STAT6, which is activated by IL-4 and IL-13, inhibit the NF-κB signalling pathway. STAT3 can lead to transcriptional activation of genes encoding pro-inflammatory cytokines including IL-6, IL-17, and IL-23.

**Figure 2 cells-12-02766-f002:**
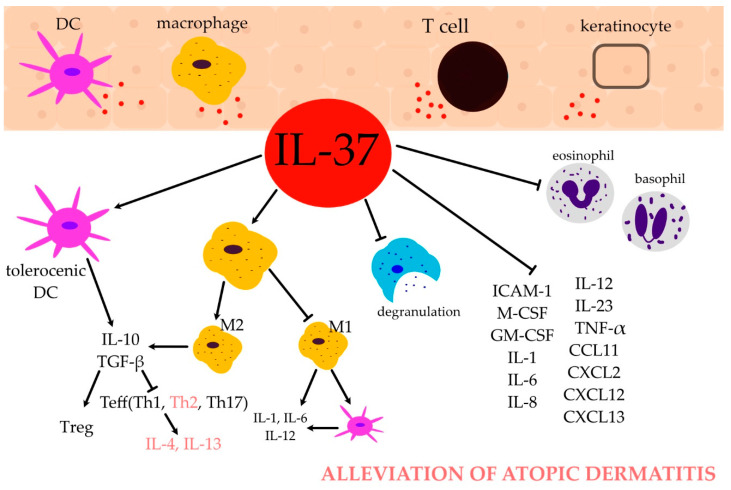
Interleukin (IL)-37 is mainly secreted by macrophages, dendritic cells (DCs), T cells, and epithelial cells (including keratinocytes). IL-37 promotes the development of tolerogenic DCs and M2 macrophages, which produce, among other anti-inflammatory agents, transforming growth factor (TGF)-β1 and IL-10, which suppress T effector (Teff) cells and promote T regulatory (Treg) cells. IL-37 impedes mast cells, eosinophils, and basophils as well. It also inhibits intracellular adhesion molecule 1 (ICAM-1), Macrophage Colony-Stimulating Factor (M-CSF), Granulocyte–Macrophage Colony-Stimulating Factor (GM-CSF), pro-inflammatory cytokines IL-1, IL-6-, IL-8, IL-12, IL-23, and TNF-α, as well as chemokines C-C Motif Chemokine Ligand 11 (CCL11), C-X-C Motif Chemokine Ligand (CXCL)2 (MIP-2), Chemokine C-C motif ligand 12 (CCL12), and CXCL13 (BCA-1)).

**Table 1 cells-12-02766-t001:** A concise overview of the primary investigations that emerge upon searching for the keywords “atopic dermatitis AND IL-37” in the PubMed database.

Country and Authors	Study Subject or Model	Most Important Findings
China 2020; Hou et al. [11]	CRISPR/Cas9 human IL-37b knock-in mice	IL-37b impeded the in vitro production of pro-inflammatory cytokines IL-6 and TNF-α, and chemokines CXCL8, CCL2, and CCL5. Il-37 enhanced LC3 conversion, an indicator of autophagy, and decreased p62, an indicator of autophagy flux, and piled inside cells when autophagy was inhibited, and the other way around. IL-37b substantially boosted Foxp3+ regulatory T cells (Treg) and IL-10, and reduced eosinophil infiltration in ear lesions. IL-37b repaired the disrupted gut microbiota diversity by modulating the intestinal metabolite-mediated autophagy mechanism.
China 2021; Hou et al. [12]	CRISPR/Cas9 human IL-37b knock-in mice or mice with direct treatment with human IL-37b antibody	IL-37b reduced thymic stromal lymphopoietin (TSLP) expression as well as the expression of TSLP receptor and basophil activation marker CD203c on basophils. IL-37 reduced the release of IL-4. Upon direct treatment with a human IL-37b antibody, AD symptoms such as ear swelling and itching were alleviated.
United States of America 2019; Guttman-Yassky et al. [13]	51 children (less than 5 years); 21 with moderate-to-severe AD with less than 6 months of disease duration; 30 did not have AD; RNA extracted from tape strips; quantitative RT-PCR	IL-37 was significantly downregulated in the skin barrier of children with AD.
Japan 2022; Tsuji et al. [14]	normal human epidermal keratinocytes	The upregulation of IL-37 in normal human epidermal keratinocytes by Tapinarof and Galactomyces ferment filtrate was blocked by the suppression of aryl hydrocarbon receptor (AHR). IL-37 increased the expression of IL-33. Tapinarof and Galactomyces ferment filtrate inhibited IL-37 expression.
United States of America 2021; Zhou et al. [15]	skin and blood samples from moderate-to-severe AD treated with topical corticosteroid and those with no topical medications applied to skin for a period of at least 1 week	A substantial decrease in IL1F7 transcripts (encoding IL-37) was observed in AD patient samples, which correlated with decreased transcript levels for important skin barrier function genes. The addition of Th2 cytokines to the skin model was sufficient to reduce epidermal IL-37 levels and develop critical characteristics of AD skin.
Norway 2020; Lossius et al. [16]	16 adult patients (5 males, 11 females; mean age 32, range 20–73); patients received standard full-body nb-UVB therapy 3×/week for 6–8 weeks with a starting dose of 0.1–0.2 J/cm^2^ and progressive dose increase; analysis of different questionnaire of AD severity	After treatment with nb-UVB, the pro-inflammatory IL-36 decreased, while the anti-inflammatory IL-37 increased.
China 2021; Hou et al. [17]	blood samples from 20 healthy control subjects (aged 14–57 years) without history of skin, allergic, or inflammatory disease, and 34 subjects (aged 11–49 years) with moderate-to-severe AD; human HaCaT keratinocytes and 3D keratinocytes	Levels of IL-37 and its receptor IL18R were significantly reduced in AD patients. A negative correlation was observed between IL-37 and involucrin, and IL-37 was shown to suppress involucrin expression in in vitro epidermal cell models.
Denmark 2022; Hu et al. [18]	30 patients with AD and 30 healthy controls; 393 skin samples from multiple anatomical regions and time points; 1.5 mm minipunch biopsies for obtaining multiple full-thickness skin samples from the same subject over time	The expression of IL-37 decreased in descending order from healthy controls to non-lesional to lesional tissues.

## Data Availability

All the data can be found in the PubMed database—https://pubmed.ncbi.nlm.nih.gov/ (accessed on 12 November 2023) or under the links cited of cited websites.

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
