# Peer review of "Potential Role of IL-37 in Atopic Dermatitis"

_cells, 2023, doi:10.3390/cells12232766_

Round 1
Reviewer 1 Report
Comments and Suggestions for Authors
The current manuscript provides an overview of the potential role of IL-37, a newly identified cytokine, in atopic dermatitis(AD). Interestingly, IL-37 is known for its anti-inflammatory properties, and it may have several potential clinical applications in treating diseases such as inflammatory or autoimmune diseases, cardiovascular diseases, and cancers. While several review articles that discussed the roles of IL-37 in AD have been published, a systemic review could be interesting. However, authors need to address the following issues:
- A brief section of the published review regarding the potential roles of IL-37 in atopic dermatitis is necessary in the Introduction.
- In Materials and Methods, a flow chart that illustrates the search methods, including, excluding criteria, will be extremely helpful, such as those from PRISMA guidelines.
- The anti-inflammatory effects of IL-37 require the co-receptors IL-18R and IL-1R8. The change in IL-18 expression and activation needs to be discussed, as IL-18 may have a significant effect on IL-38.
- Interpretation of the anti-inflammatory signals with signaling molecules needs to be careful (Figure 1). For example, STAT3 can lead to transcriptional activation of genes encoding proinflammatory cytokines, including IL-6, IL-17, and IL-23.
- Despite the preclinical studies or animal studies that revealed the role of IL-37 in AD, the IL-37 expression may not change in all AD patients (Line 227-228). The potential reasons should be discussed.
- Although the manuscript described the interaction of many immune cells and IL-38, IgE or B cells could be necessary in some subsets of the Ad patients. A brief discussion of the relationship with or regulation of B cells (plasma cells) will be helpful.
- Many AD patients have filaggrin gene mutations, particularly null mutations. Thus, treating these patients with IL-37 by enhancing the expression of FLG genes may not work as expected. This has to be included in the discussion.
- Minor language/spelling issues, such as those in Line 47 and Line 202, have to be fixed.
Reviewer 2 Report
Comments and Suggestions for Authors
This review article entitled “Potential Role of IL-37 in Atopic Dermatitis” gives us insights into a role of IL-37 in the inflammatory response.
IL-37 seems to play broad anti-inflammatory roles in various diseases, but its detailed function in each disease remains elusive.
There are a few questions to be addressed.
The expression of IL-37 is elevated in response to inflammatory stimuli, but AD patients have impaired production of IL-37.
What is the most possible cause of IL-37 downregulation in AD?
It makes me wonder why IL-37 is secreted by a broad range of cell types including macrophages, dendritic cells (DCs), T cells, and epithelial cells etc.
Does the source of IL-37 vary depending on the inflammatory organs?
Reviewer 3 Report
Comments and Suggestions for Authors
The manuscript is well written
Table 1 must be cited in the text of the manuscript. Some references are too old to be cited in the manuscript. Please use references until 10 years.
Round 2
Reviewer 1 Report
Comments and Suggestions for Authors
The authors have addressed the questions raised in the previous version, and the manuscript has been significantly improved.
